# Comparative Study on Nutrition and Lifestyle of Information Technology Workers from Romania before and during COVID-19 Pandemic

**DOI:** 10.3390/nu14061202

**Published:** 2022-03-12

**Authors:** Bogdana Adriana Nasui, Andreea Toth, Codruta Alina Popescu, Ovidiu Nicolae Penes, Valentin Nicolae Varlas, Rodica Ana Ungur, Nina Ciuciuc, Cristina Alina Silaghi, Horatiu Silaghi, Anca Lucia Pop

**Affiliations:** 1Department of Community Health, “Iuliu Hatieganu” University of Medicine and Pharmacy, 6 Louis Pasteur Street, 400349 Cluj-Napoca, Romania; adriana.nasui@umfcluj.ro (B.A.N.); nina.ciuciuc@umfcluj.ro (N.C.); 2Institute of Urology and Kidney Transplant, 4-6 Clinicilor Street, 400006 Cluj-Napoca, Romania; deea.toth@yahoo.com; 3Department of Practical Abilities—Human Sciences, “Iuliu Hatieganu” University of Medicine and Pharmacy, 6 Louis Pasteur Street, 400349 Cluj-Napoca, Romania; 4Department of General Medicine, “Carol Davila” University of Medicine and Pharmacy, 37 Dionisie Lupu Street, 020021 Bucharest, Romania; 5Department of Obstetrics and Gynaecology, Filantropia Clinical Hospital, 050474 Bucharest, Romania; valentin.varlas@umfcd.ro; 6Department of Rehabilitation, Faculty of General Medicine, “Iuliu-Hatieganu” University of Medicine and Pharmacy, 8 Victor Babes Street, 400012 Cluj-Napoca, Romania; rodica.ungur@umfcluj.ro; 7Department of Endocrinology, “Iuliu Hatieganu” University of Medicine and Pharmacy Cluj-Napoca, Victor Babes Street 8, 400012 Cluj-Napoca, Romania; alina.silaghi@umfcluj.ro; 8Department of Surgery V, “Iuliu Hatieganu” University of Medicine and Pharmacy Cluj-Napoca, Victor Babes Street 8, 400012 Cluj-Napoca, Romania; horatiu.silaghi@umfcluj.ro; 9Department of Clinical Laboratory, Food Safety, “Carol Davila” University of Medicine and Pharmacy, Traian Vuia Street, 020945 Bucharest, Romania; anca.pop@umfcd.ro

**Keywords:** IT workers, pandemic, pre-pandemic, lifestyle, diet, screen time, physical activity, stress, sleep

## Abstract

The study aimed to evaluate lifestyle factors among Information Technology (IT) workers from Romania before and during the pandemic. We used an online applied questionnaire, filled in by 1638 respondents, that assessed nutrition status-Body Mass Index (BMI), weight and diet change, physical activity, alcohol consumption, number of hours working in front of the computer, stress, and sleep. Multivariate logistic regression was used to establish the lifestyle factors that lead to weight gain. Although the level of physical activity (PA) was low before the pandemic, the results of our study showed a further decrease in physical activity. In total, 61.1% of men and 71.1% of women performed PA for less than 30 min per day. Weight gain was reported in 50.5% of men and 45.3% of women (mean weight gain was 5.11 ± 3.52 kg) as a result of increased screen time (with a mean of 3.52 ± 4.29 for females and 3.05 ± 2.09 for males, *p* > 0.05 h per day) and the reported changes in diet. Despite the popularity of home-cooked foods, the intake of vegetables and fruit remained low. The quality of sleep was poor for 55.7% of the respondents. Public health and corporation policies are required to encourage a healthy lifestyle and avoid chronic diseases.

## 1. Introduction

It is well known that Information Technology (IT) workers are a category of the population with a highly sedentary lifestyle [1]. Sedentary behavior will lead to different pathologies (obesity, cardiovascular diseases, diabetes, cancers) [1]. Sedentary lifestyles are spreading worldwide because of increased occupational sedentary behaviors such as office work and increased penetration of television and video devices [2].

The invasive nature of technology and social media can induce technostress in workers [3,4]. Technostress refers to the stress that people experience due to the use of technology and the demands related to technology’s use [5,6,7]. In addition, continuous online meetings can be exhausting, and multitasking and concentration problems can occur, leading to fatigue, exhaustion, stress, and burnout [8]. On the other hand, social distancing due to the COVID-19 pandemic may have impacted the basic human need for social belonging and increased the likelihood of loneliness. Loneliness can be seen as an interpersonal stressor that gives rise, for instance, to negative emotions and physiological reactivity, which is conducive to the nervous system’s unhealthy activity. In line with this, adults experiencing loneliness are also characterized by higher levels of anxiety, negative mood, and stress [9,10]. 

A sedentary lifestyle has also been found to increase the risk of depression and anxiety. 

In many countries, governments have adopted restrictive measures such as quarantine, social distancing, the suspension of any social event, and the closure of schools, universities, gyms, sports centres, swimming pools, and public parks. All these restrictions can reduce public access to physical activity opportunities [11]. In addition, teleworking was encouraged, allowing individuals to work from their homes.

Studies from the literature showed that workers with a balanced internet use (worktime) had a higher perceived quality of life compared to individuals with a little or large amount of internet use for work. These results are largely consistent when taking negative feelings, perceived stress, smoking status, and alcohol consumption status into account [12]. 

Sedentary behaviour is found to be associated with an increased risk of insomnia and sleep disturbances [13]. A vicious cycle between sleep problems and physical activity may exist, i.e., sleep problems may lead to less physical activity, and less physical activity may worsen sleep problems [14].

The Corona Virus 19 (COVID-19) pandemic has infected millions worldwide and impacted many more. On March 11, the World Health Organization (WHO) declared COVID-19 as a pandemic [15]. The virus was confirmed to have reached Romania on 26 February 2020, when the first case in Gorj County was confirmed. On 8 March the Head of the Department of Emergency Situation announced a ban on all indoor and outdoor activities involving more than 1.000 people. On March 16, President Klaus Iohannis announced his decision to decree a state of emergency in Romania starting on 16 March and on 24 March the government announced Military Ordinance No.3, instituting a national lockdown. On 1 October 2021, the National Institute of Public Health had reported around 1,210,000 cases, 1,100,000 recoveries, and 37,200 COVID-19-related deaths [16].

Socioeconomic changes that occurred in the last year have forced companies to be more flexible and offer their employees the possibility to work remotely as an exception. Remote working began during the pandemic lockdown and continued throughout the rest of the COVID-19 pandemic. Social isolation increases sedentary behavior, and screen time could lead to changes in nutrition habits and weight gain. The restaurants and bars were also closed, leaving the options for takeaway or forcing people to cook at home.

Stress is related to higher energy intake, food craving, unhealthy dietary patterns higher in fat and sugar, and higher alcohol consumption, leading to weight gain and increased obesity. Drinking alcohol is associated with various short and long-term health risks, including motor vehicle crashes, violence, sexual risks, blood pressure, and cancers (e.g., breast cancer) [17,18,19]. Moderate alcohol intake has protective health benefits, reducing the risk of heart disease. 

There are only a few European and US studies, two briefly addressing health and lifestyle factors and the remainder evaluating stress/burnout [1,20]. There are no Romanian studies assessing IT workers’ health behavior and lifestyle factors to our best knowledge. The present study aimed to assess the lifestyle of IT workers before the pandemic and during the pandemic, as dietary behaviors, nutrition status, physical activity, sleep, working hours, and stress. 

## 2. Materials and Methods

### 2.1. Study Sample and Data Collection

To achieve the objectives of the study, we used a cross-sectional survey. The questionnaires were administered online using Google Docs. The link to the questionnaire was distributed using the emails taken from the websites of different IT institutions from all over the country; the IT companies who agreed to participate shared the link on their internal groups. To obtain more participants, we also emailed the link to friends working in IT with the same request to share the questionnaire. Inclusion criteria were: individuals working in the IT domain and aged over 18 years because all these participants graduated at least high school, and the majority had a university degree. One thousand six hundred eighty-three persons from Information Technology (IT) participated in this survey. From these 715 were selected before the pandemic and 923 were selected during the pandemic with SARS-CoV-2. We collected the data for the pre-pandemic sample in September–November 2019 and for the pandemic respondents from October 2020 to January 2021.

### 2.2. Questionnaire and Variables

We used a self-administered questionnaire. The questionnaire was elaborated/developed based on previous studies and based on international recommendations for a healthy lifestyle [21,22]. We tested and re-tested the questionnaire on a sample of 30 respondents and calculated Spearman’s correlation coefficient to assess the reliability (r = 0.766). The time required to fill the questionnaire in the survey was 15–20 min.

The questionnaire included the following:Demographic data: sex, age, studies, marital status;Number of hours of working in front of the computer (screen time) or other sedentary behaviors like driving a car, watching television, using other electronic devices. During the pandemic, the questionnaire comprised additional questions like “did you work more hours during the pandemic” “did you work at home” (remote work)?

Diet was assessed as follows:
c.Diet as the numbers of meals per day, fast food consumption, vegetables, and fruit consumption. In the case of vegetables and fruits, we investigated how many portions were eaten per day. A portion of fruits or vegetables was considered to be 80 g of a medium-size fruit or a 1/2 cup of chopped/cooked vegetables, 30 g of dried fruits, or 150 mL of fruit or vegetable juice.

The respondents were asked if they changed their diet during the pandemic, with possible answers yes or no. The questionnaire investigated if the food was homemade during the pandemic (“The food was homemade? - possible answers yes or no”);

d.Alcohol consumption.

The questionnaire also estimated alcohol consumption as the number of drinks per week. We defined a standard drink as 500 mL beer with 5% alcohol or 150 mL wine with 12.5% alcohol or 50 mL of distillate beverage (40% alcohol); 

e.Physical activity

Physical activity was assessed as the duration and frequency, and the place where the physical activity was performed. As an alternative, the number of steps recorded by devices for those who had bracelets, fitness watches, or other devices. The answers were divided into 3000–4000 steps, between 4001–8999 steps, 9000–10,000, and over 10,000 steps;

f.Sleep was assessed as quantity (How many days per week do you sleep less than six hours?) and as quality (“How do you wake up in the morning? Tired or rested”);g.Stress was investigated by self-assessment, using the following question and answer categories: low, moderate, or high. The study also investigated the social need of meeting other persons using a scale from 1 to 10 points;h.Weight and height were reported as self-assessment. We calculated the body mass index using the formula BMI = Weight (kg)/Height^2^ (m^2^). We divided the sample according to WHO guidelines as: underweight –BMI<18.5; normal weight: BMI = 18.5–24.9; overweight was defined as BMI= 25–29 kg/m^2^ and obesity as BMI >30 kg/m^2^. The questionnaire also investigated the weight gain or weight loss of the respondents.

### 2.3. Statistical Analysis

Data were statistically analyzed using the Statistical Package for the Social Sciences, version 20. Descriptive statistics (means and standard deviations for continuous variables or frequencies and percentages for categorical variables) were calculated. A chi-square (χ^2^) statistical test was conducted to examine the significant differences for categorical variables between two periods of time (before and during the COVID-19 outbreak). We used ANOVA to compare the respondent’s weight gain depending on their Body Mass Index. 

Multiple regression was run to predict the weight gain due to lifestyle change after the COVID-19 outbreak from gender, age, BMI, diet change, and more hours of work in front of the computer. All reported *p* values were 2-tailed, and *p* < 0.05 was considered to be statistically significant.

### 2.4. Ethical Issues

The study was approved by the Bioethics Committee of the University of Medicine and Pharmacy Iuliu Hatieganu, Cluj-Napoca, No 254/2019. No personal data were collected, and the questionnaires were anonymous and voluntary. The participants were informed about the aim of the study, and by completing the questionnaire, they agreed to participate in the study.

## 3. Results

The mean age of the participants was 26.58 ± 6.33 years for the respondents during the pandemic period and 27.02 ± 6.42 (*p* = 0.06) in the pre-pandemic period. Of the investigated sample, 57.4% (941) were males, and 42.6% (697) were females. There was no statistically significant distribution depending on sex between the samples of the two periods (*p* = 0.078).

The results of the present study revealed that 11.5% (*n* = 188) of respondents graduated high school, 56.2% (*n* = 921) graduated from university, 31.8% (*n* = 521) had a Master’s degree and 0.5% (*n* = 8) had PhD degree. Regarding marital status, 20.3% (*n* = 333) were married, 46.3% (*n* = 759) were involved in a relationship, 32.1% (*n* = 526) were single, and 1.2% (*n* = 20) were divorced.

### 3.1. Assessment of Nutrition Status and Weight Gain

Regarding the body mass index, the present study revealed that about a half of males 53.4% of the pre-pandemic sample and 50.4% from the pandemic sample, were normal weight. However, a large percentage of males were overweight or obese, 44.6% and 47.9%, respectively. The majority of the females were normal weight (69.9% both samples, pre-and pandemic), and 21.1% and 21.8% were overweight or obese, respectively (Table 1). The mean BMI was 26.81 ± 6.17 kg/m^2^, higher in males (25.37 ± 4.03 kg/m^2^) than females (22.58 ± 3.88 kg/m^2^) (*p* = 0.59).

Regarding the weight gain during the pandemic, the vast majority of the men (50.5%, *n* = 277) and almost half of the women (45.3%, *n* = 170) reported a weight gain (*p* = 0.123). The mean weight gain was 5.16 ± 4.07 kg for males and 4.14 ± 3.97 kg for females (*p* = 0.006) (mean weight gain of the sample was 5.11 ± 3.52 kg. 27% of the respondents reported a weight loss. The weight loss was 4.29 ± 4.12 kg in males and 3.84 ± 2.98 in females (*p* = 0.003) (mean weight loss 4.01 ± 4.17 kg). We calculated the respondents’ weight gain depending on their body mass index. The results revealed that IT workers who were overweight or obese gained more weight than normal weight workers (Table 2).

### 3.2. Diet and Changes

Diet was estimated as the number of meals consumed per day, intake of vegetables and fruit, and fast-food intake. In addition, the respondents were asked if they changed their diet during the pandemic.

The majority of the computer workers ate three meals per day as recommended. However, the study results showed that a significant percentage of respondents had only two meals per day, a higher percentage in the pandemic period (Table 3).

The main meal was eaten at home, during the pandemic period, as a result of remote work. However, in the pre-pandemic period, a significant percentage of computer workers, especially men (48.3%), ate the main meal in town (Figure 1). 

Vegetables were eaten in small amounts in both periods. Apparently, the results of our study showed no increase in vegetable intake (Table 4). Fruits were consumed in a higher amount than vegetables. Most of the workers ate 1–2 fruits per day regardless of the investigated period.

The present study revealed a higher fast-food intake among males than females (*p* < 0.001) during the pre-pandemic period. A high percentage of males consumed fast food 2 or 3 times per week. During the pandemic, the results of our study showed an increase in fast food consumption both in males and females (*p* = 0.228) (Table 5).

Furthermore, the respondents were asked if they changed their diet during the pandemic, to which 52.9% (*n* = 290) of males and 62.4% (*n* = 234) of females IT workers (*p* = 0.005) answered affirmatively. The majority of the pandemic sample of computer workers (80%, *n* = 740) declared that the food was homemade, and the rest (19.8%, *n* = 183) of the respondents answered that the main meal was ordered. 

### 3.3. Alcohol Consumption

We estimated alcohol consumption. Our results showed that alcohol consumption decreased during the pandemic. The majority of men (77.6%) did not drink during the pandemic, and 9.9% drank 3–5 times per week in comparison with 24.4% in the pre-pandemic period (Table 6). Most of the females were abstainers (84.8% in the pre-pandemic period versus 87.5% during the pandemic period)

### 3.4. Physical Activity

Males performed a higher amount of physical activity than females in both the studied periods (pre-pandemic and pandemic). The study revealed a high percentage of males (30.5% during the pre- and 61.2% during the pandemic) that were not meeting the physical activity recommendations. Females had a higher level of sedentary behaviors, both in pre-pandemic (71.1%) and in remote working periods (70.45%) (Table 7).

In addition, the study estimated the level of physical activity depending on the number of steps in those who had devices or applications on the smartphone. In the pandemic period, the study revealed a low interest in step counting, a higher percentage of respondents did not/have or did not count their steps, and a decrease in the number of the steps, both in males and females (Table 8).

### 3.5. Hours Spent in Front of the Computer (Screen Time)

The results of our study showed that over half of the males (52.7%) and females (52.5%) IT workers stayed more than 8 h in front of the computer or driving a car before the pandemic. During the pandemic, the percentage increased, 84.7% of males and 85.1% of females were involved in teleworking and sedentary behaviors (Figure 2).

In addition, during the pandemic, the present study results showed that IT workers spent more hours in front of computers, more males than females (*p* < 0.001). In total, 27.9 % (*n* = 153) of the male and 17.6% (*n* = 66) of the female respondents reported that they had exceeded the regular program during this pandemic period, with an average of hours of 3.52 ± 4.29 for females and of 3.05 ± 2.09 for males, (*p* > 0.05).

Teleworking was the method of work applied during the pandemic. The majority of the respondents, 90.5% (*n* = 496) males and 89.1% (*n* = 334) females, worked from home during the pandemic (*p* = 0.505).

### 3.6. Stress and Social Isolation

Social isolation was estimated during the pandemic period. The majority of the male respondents - 89.4%, and 90.4% of the female respondents (*p* = 0.659) declared that they needed to see their friends during the pandemic. On a scale from 1 to 10, males suffered less than females from social isolation (5.29 ± 2.37 in males vs. 6.01 ± 2.17 in females, *p* < 0.001). 

Regarding the self-assessment of stress due to the job assignment, the majority of the respondents perceived their stress as moderate during both periods (Table 9).

### 3.7. Sleep

The results of our study showed that only 34.5% of the respondents before the pandemic period and 35.4% during the pandemic had the recommended amount of sleep. More workers slept less than six hours during the pandemic, at least once per week. In addition, more than half of the IT workers felt tired in the morning, meaning they had poor sleep quality (Table 10).

### 3.8. Factors Associated with Weight Gain

We ran a multiple regression to predict the weight gain of the computer workers during the pandemic period depending on gender, age, BMI, diet change, and more hours of work in front of the computer. These variables significantly predicted the weight gain F (5.917) = 42.965, *p* < 0.000, R^2^ = 42.965. According to our results, age, BMI, diet change, and more hours of work were statistically significantly associated with the weight gain of the respondents (*p* < 0.05) (Table 11).

## 4. Discussion

The study aimed to estimate the lifestyle of computer workers, such as the number of hours spent on work using the computers, nutrition status, physical activity, diet, sleep, perceived stress, stress isolation before the pre-pandemic period and during the pandemic period. Sedentary lifestyles are spreading worldwide because of a lack of available spaces for exercise, increased occupational sedentary behaviors such as office work, and the increased penetration of television and video devices. Consequently, associated health problems are on the rise. A sedentary lifestyle increases all-cause mortality and the risks of cardiovascular diseases (CVD), diabetes mellitus (DM), hypertension (HTN), and cancers (breast, colon, colorectal, endometrial, and epithelial ovarian cancer). This has been consistently documented in the literature [1,23]. The present study showed that more men (50.5%) during the pandemic than women gained weight (45.3%). The results are consistent with other studies that showed an increase in body weight during the pandemic [24]. The weight gain was associated with an increase in screen time and decreased physical activity [25]. Less IT workers in the present study lost weight (4.01 ± 4.17 kg). Other studies from literature evidenced that weight loss during the pandemic was associated with healthy eating changes (increased consumption of vegetables, fruits, cereals, fish, water, and a decrease in the intake of discretionary foods during the quarantine compared to other respondents [26]. Our study showed a low intake of vegetables and fruits and an increase in fast food intake. However, future studies are needed to better estimate other food groups’ consumption during the pandemic and the relationship with the nutrition status among Romanian IT workers.

Studies from the literature showed that the pandemic had a negative effect on physical activity by decreasing it and increasing screen time [27]. The present study revealed that 84.7% of male and 85.1% of female IT workers during the pandemic spent more than 8 h in front of the computers in comparison with 52.7% of the males and 52.5% of the females in the pre-pandemic period. The multivariate analyses showed that the increased time spent in front of the computers was a risk factor for weight gain. 

As our results showed, the reduced amount of physical activity among IT workers before the pandemic was also recorded during the pandemic. We estimated the physical activity as time spent; counting the steps; the percentage of the sample; comparison with pre-pandemic IT workers. These results are similar to other studies that showed a decreased level of physical activity [28,29]. Besides the reduction in PA, the study revealed an increased time in sedentary behaviors, like working at the computer.

The WHO/FAO report recommends 400 g edible fruit and vegetables per day as a population-wide intake goal for the prevention of non-communicable diseases (NCDs) and the prevention and alleviation of several micronutrient deficiencies. This translates to roughly five portions per day (two portions of fruits and three portions of vegetables (potatoes, sweet potatoes, cassava, and other starchy roots are not classified as fruits or vegetables) [30]. In European countries, the level of fruit and vegetable intake is still poor. The results of our study showed that over 70% of the IT workers in the pre-pandemic period consumed 1–2 portions of fruits (72.8% for males and 72.1% for females). However, during the pandemic period, the fruit intake was lower (51% for males and 56% for females). The present study revealed that vegetable consumption was insufficient during both periods (44.3 vs. 39.6% for males and 46.6% vs. 44.5% for females). However, these intakes were higher than in other young Romanian populations, as previous studies showed [31]. Other studies undertaken during the pandemic on different European populations showed either an increase in vegetable and fruit intake (18.5% and 15.2%) [32] or a decrease in these food groups. 

The intake of fast-food increased during the pandemic period, consumption twice per week, from 15.7% (in pre-pandemic) to 20.0% (in pandemic) of the respondents (*p* < 0.001). Data from the literature are inconsistent. Some studies from the literature showed that fast-food consumption was increased [31]. Most studies however showed a decrease in fast-food intake [33]. Fast food, processed food, and junk food contribute to obesity, diabetes, heart attacks, strokes, dementia, and cancer. However, many do not realize the strong causative role of an unhealthy diet in mental illness [34]. 

Both alcohol consumption and smoking can potentially increase vulnerability to SARS-CoV-2 infection and worsen the clinical course of COVID-19. Chronic alcohol exposure has a complex and adverse effect on host response [35]. Stress is a prominent risk factor for the onset and maintenance of alcohol misuse. The reported stress among computer workers was moderate in both studied periods. The results of our study showed that the majority of IT workers did not consume alcohol, before and during the pandemic. A decrease in alcohol consumption was seen in the men, 24.4% of male IT workers drank 3–5 times per week before the pandemic versus 9.9% of male IT workers during the pandemic period (*p* = 0.004). These results are inconsistent with other studies that showed an increase in alcohol consumption, with a higher tendency to drink among alcohol addicts [24]. Another study revealed a decrease in alcohol consumption associated with the Mediterranean diet during the lockdown [36,37]. 

Poor sleep quality may lead to weight gain, worsened mood, increased stress, depression and anxiety, daytime fatigue, and poor concentration [19,38]. The recommended amount of sleep per night is 7–9 h for adults. The present study revealed that 33.6 % before the pandemic and 41.7% IT workers during the pandemic slept once or twice per week less than six hours, and over a half reported poor quality of sleep during both periods (they felt tired in the morning). Studies from the literature showed that excessive screen time could lead to poor sleep quality [39,40].

The present study was the first research that assessed the lifestyle of computer workers from Romania and is an essential study in a lacking data field; consequently, conclusions can be drawn.

Given the fact of the increased demand for computer workers, the study emphasized the need for the implementation of educational and public health measures among this population group to prevent chronic diseases. Public health policies and benefits offered by companies (flexible hours, gym memberships, financial support, catering contracts with nutritional restaurants, healthy “in office” kitchens) are necessary to encourage an active lifestyle. An active lifestyle can prevent weight gain and improve sleep and mood. Dietary changes are recommended, increasing the intake of vegetables, seeds, and fruits, with the assistance of professional nutritional support (training and nutritional specialists’ appointments). 

### Limitation

The main limitation of the study was the self-reported questionnaire. Given the data collection period, during the pandemic, and the population group, we considered remote data collection on specific professional social network sites of computer workers as feasible. Due to this data collection method, we succeeded in having a large sample of respondents. Another limitation of our study was that we could not reach the same sample of IT workers during the pandemic as before the pandemic (the questionnaires were anonymous). However, the two samples were comparable, without statistical differences regarding the age and sex distribution. Regardless of the interest of IT workers in counting their steps, future studies are necessary to use an objective measurement of physical activity (accelerometer, pedometer) and with standardized tools for stress. Future studies are needed to understand better the outcomes of home confinement during the pandemic.

## 5. Conclusions

The present study investigated the lifestyle of computer workers from Romania, an important population group with high sedentary behaviors. As a consequence of the pandemic, teleworking further increased screen time and decreased physical activity, leading to weight gain in some of the respondents. The study also revealed poor sleep quality among the majority of respondents, with at least one weeknight with less than 6 h of sleep.

Regardless of the significant number of working hours in front of computers, most IT workers perceived their work stress as only moderate. Moreover, our study revealed that almost all workers suffered from social isolation during the pandemic, especially women.

In addition, diet changes have occurred, with a considerable growth in the consumption of homemade food; however, the intake of vegetables and fruit remained low. Public health and corporation policies and interventions are required to encourage a healthy lifestyle and avoid workers’ chronic diseases. 

## Figures and Tables

**Figure 1 nutrients-14-01202-f001:**
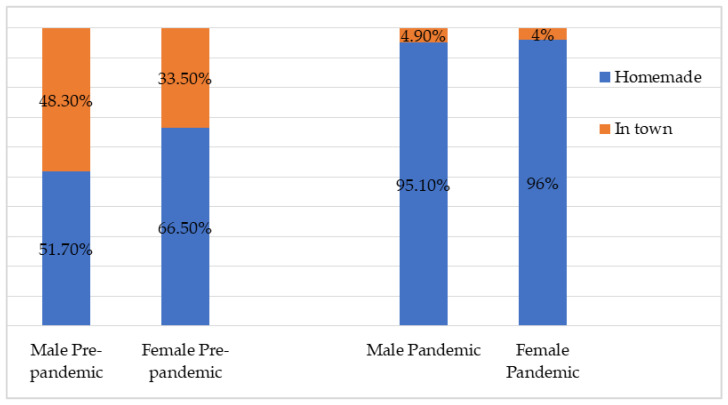
Where did you eat the main meal.

**Figure 2 nutrients-14-01202-f002:**
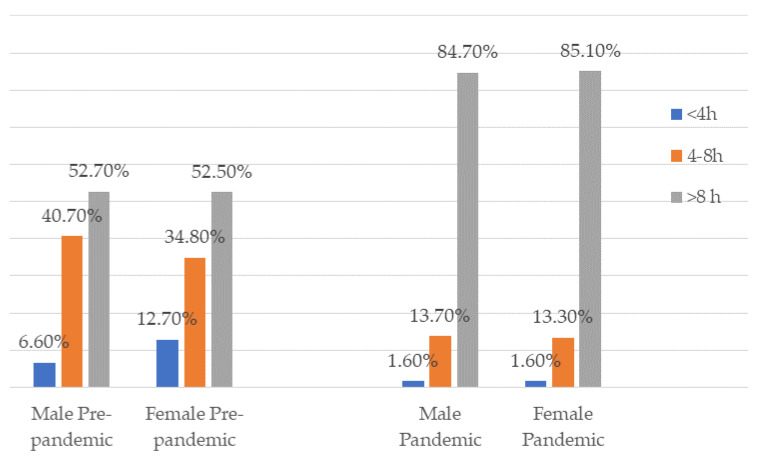
The number of hours spent in front of the computer/driving a car/watching TV.

**Table 1 nutrients-14-01202-t001:** Body mass index categories for IT workers.

Variable BMI	Male Pre-Pandemic 2019 No (%)	Female Pre-Pandemic 2019 No (%)	*p* Value *	Male Pandemic 2020	Female Pandemic 2020	*p* Value *
Underweight	8 (2)	29 (9)	0.0001	9 (1.6)	31 (8.3)	0.0001
Normal weight	210 (53.4)	225 (69.9)		276 (50.4)	262 (69.9)	
Overweight	137 (34.9)	54 (16.8)		203 (37)	65 (17.3)	
Obesity	38 (9.7)	14 (4.3)		60 (10.9)	17 (4.5)	

* *p* < 0.05 was considered statistically significant; Chi square.

**Table 2 nutrients-14-01202-t002:** The weight gain of the respondents depending on the BMI.

Obesity Class (BMI)	Mean ± SD (kg)	No (%)	*p*-Value *
Underweight	5.83 ± 5.49	8 (1.79)	*p* < 0.001
Normal weight	5.85 ± 2.94	218 (48.80)	
Overweight	6.81 ± 3.56	170 (38.00)	
Obesity	8.24 ± 4.80	51 (11.41)	

* *p* < 0.05 was considered statistically significant, ANOVA.

**Table 3 nutrients-14-01202-t003:** The number of meals per day.

No. of Meals/Day	Male Pre-Pandemic No (%)	Female Pre-Pandemic No (%)	*p*-Value *	Male Pandemic No (%)	Female Pandemic No (%)	*p*-Value *
1	5 (1.3)	5 (1.6)	0.390	9 (1.6)	8 (2.1)	0.510
2	88 (22.4)	84 (26.1)		169 (30.8)	129 (34.4)	
3	224 (57)	179 (55.6)		285 (52)	179 (47.7)	
4	66 (16.8)	41 (12.7)		69 (12.6)	42 (11.2)	
5	5 (1.3)	9 (2.8)		13 (2.4)	15 (4.0)	
6	5 (1.3)	4 (1.2)		3 (0.5)	2 (0.5)	

* *p* < 0.05 was considered statistically significant; Chi Square.

**Table 4 nutrients-14-01202-t004:** The number of portions of vegetables and fruits per day.

Number of Portions	Male Pre-Pandemic No (%)	Female Pre-Pandemic No (%)	*p* Value *	Male Pandemic No (%)	Female Pandemic No (%)	*p* Value *
Vegetables						
0	15 (3.8)	15 (4.7)	0.310	121 (22.1)	65 (17.4)	0.206
1	187 (47.6)	135 (41.9)		180 (32.8)	124 (33.1)	
2	125 (31.8)	115 (35.7)		146 (26.6)	126 (33.6)	
3	49 (12.5)	35 (10.9)		69 (12.6)	41 (10.9)	
>3	17 (4.3)	22 (6.8)		32 (5.8)	19 (5.1)	
Fruit						
None	53 (13.5)	29 (9.0)	0.005	202 (36.9)	131 (34.9)	0.474
1	204 (51.9)	139 (43.2)		157 (28.6)	125 (33.3)	
2	82 (20.9)	93 (28.9)		123 (22.4)	85 (22.7)	
3	45 (11.5)	45 (14.0)		34 (6.2)	21 (5.6)	
>3	9 (2.3)	16 (5.0)		32 (5.8)	13 (3.5)	

* *p* < 0.05 was considered statistically significant; Chi Square.

**Table 5 nutrients-14-01202-t005:** The frequency of fast-food consumption per week.

	Male Pre-Pandemic No (%)	Female Pre-Pandemic No (%)	*p*-Value *	Male Pandemic No (%)	Female Pandemic No (%)	*p*-Value *
<1/week	145 (36.9)	210 (65.2)	0.0001	244 (44.6)	187 (49.9)	0.228
1/week	93 (23.7)	47 (14.6)		109 (19.9)	81 (21.6)	
2/week	70 (17.8)	42 (13.0)		117 (21.4)	68 (18.1)	
3/week	68 (17.3)	17 (5.3)		57 (10.4)	24 (6.4)	
4/week	10 (2.5)	2 (0.6)		9 (1.6)	7 (1.9)	
>=5/week	7 (1.8)	4 (1.2)		12 (2.2)	8 (2.1)	

* *p* < 0.05 was considered statistically significant; Chi-Square.

**Table 6 nutrients-14-01202-t006:** Alcohol consumption per week.

Frequency of Alcohol Consumption	Male Pre-Pandemic No (%)	Female Pre-Pandemic No (%)	*p*-Value *	Male Pandemic No (%)	Female Pandemic No (%)	*p*-Value *
No or/<1 week.	213 (54.2)	273 (84.8)	<0.001	425 (77.6)	328 (87.5)	<0.001
3–5 times/week	96 (24.4)	30 (9.3)		54 (9.9)	33 (8.8)	
5–7 times/week	57 (14.5)	12 (3.7)		42 (7.7)	8 (2.1)	
>7 times/week	27 (6.9)	7 (2.2)		27 (4.9)	6 (1.6)	

* *p* < 0.05 was considered statistically significant; Chi-Square.

**Table 7 nutrients-14-01202-t007:** Physical activity duration per day.

PA Minutes/Day	Male Pre-Pandemic No (%)	Female Pre-Pandemic No (%)	*p* Value *	Male Pandemic No (%)	Female Pandemic No (%)	*p* Value *
Not at all	5 (1.3)	3 (0.9)	0.005	81 (14.8)	71 (18.9)	0.001
10 min	61 (15.5)	58 (18.0)		150 (27.4)	97 (25.9)	
20 de minutes	54 (13.7)	62 (19.3)		39 (7.1)	52 (13.9)	
30 de minutes	107 (27.2)	106 (32.9)		65 (11.9)	44 (11.7)	
>30 de minutes	166 (42.2)	93 (28.9)		213 (38.9)	111 (29.6)	

* *p* < 0.05 was considered statistically significant; Chi-Square.

**Table 8 nutrients-14-01202-t008:** Number of steps per day, (for those who have a smartwatch fitness bracelet).

No of Steps/Day	Male Pre-Pandemic No (%)	Female Pre-Pandemic No (%)	*p*-Value *	Male Pandemic No (%)	Female Pandemic No (%)	*p*-Value *
No/I don’t have	81 (20.6)	86 (26.7)	0.085	229 (41.8)	137 (36.5)	0.0001
<3000	0	0		65 (11.9)	95 (25.3)	
3000–4000	54 (13.7)	55 (17.1)		65 (11.9)	57 (15.2)	
4001–8999	141 (35.9)	109 (33.9)		124 (22.6)	66 (17.6)	
9000–10,000	72 (18.3)	42 (13.0)		38 (6.9)	12 (3.2)	
>10,000	45 (11.5)	30 (9.3)		27 (4.9)	8 (2.1)	

* *p* < 0.05 was considered statistically significant; Chi-Square.

**Table 9 nutrients-14-01202-t009:** Stress assessment.

Stress	Pre-Pandemic Period No (%)	Pandemic Period No (%)	*p* Value *
Low stress	144 (20.1)	188 (20.4)	0.335
Moderate stress	511 (71.5)	638 (69.1)	
High stress	60 (8.4)	97 (10.5)	

* *p* < 0.05 was considered statistically significant; Chi-Square.

**Table 10 nutrients-14-01202-t010:** Sleep quantity and quality.

How Often Do You Sleep Less Than Six Hours	Pre-Pandemic Period No (%)	Pandemic Period No (%)	*p*-Value *
Never	247 (34.5)	327 (35.4)	<0.001
One/week	97 (13.6)	200 (21.7)	
2/week	143 (20)	185 (20)	
3/week	109 (15.2)	95 (10.3)	
4/week	49 (6.9)	48 (5.2)	
More than 4/week	70 (9.8)	68 (7.4)	
How did you feel in the morning?			
Rested	308 (43.1)	418 (45.3)	0.200
Tired	407 (56.9)	505 (54.7)	

* *p* < 0.05 was considered statistically significant; Chi-Square.

**Table 11 nutrients-14-01202-t011:** Multiple regression regarding the weight gain during the pandemic.

Coefficients							
Model	Unstandardized Coefficients		Standardized Coefficients	t	Sig.	95.0% Confidence Interval for B	
	B	Std. Error	Beta			Lower Bound	Upper Bound
(Constant)	−6.521	0.791		−8.241	<0.001	−8.074	−4.968
More hours of work in front of the computer	0.846	0.270	0.094	3.130	0.002	0.316	1.377
Diet change	0.679	0.233	0.088	2.910	0.004	0.221	1.136
Gender	0.032	0.247	0.004	0.128	0.898	−0.454	0.517
Age	−0.051	0.019	−0.085	−2.719	0.007	−0.088	−0.014
BMI	0.388	0.030	0.427	12.978	<0.001	0.329	0.447

## Data Availability

Not applicable.

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
