# Peer review of "Comparative Study on Nutrition and Lifestyle of Information Technology Workers from Romania before and during COVID-19 Pandemic"

_nutrients, 2022, doi:10.3390/nu14061202_

Round 1
Reviewer 1 Report
First of all, we would like to thank the authors for the submitted manuscript, however, and after the first review, certain flaws are detected that must be improved in order to consider its acceptance. It is an interesting topic evaluating lifestyle factors among Romanian information technology (IT) workers before and during the pandemic.
In the Methodology section:
- It would be necessary to indicate the type/design of the study (for example, a cross-sectional, quantitative, descriptive, comparative study...
- Authors are urged to move the ethical considerations section to the end of the material and method section.
- Could the authors indicate which online platform the link was sent to the participants or indicate if it was through email or social networks?
- Was the sample obtained through the intentional or snowball method? indicate it
In results:
- The tables must all be presented in the same format in terms of font. On the other hand, adjust them so that they are not cut between pages and if this is not possible, repeat the header of the same.
- In Figure 1, on line 217, Delete the question mark
- Review the text on line 291 (a: Dependent variable: weight gain.) What does it refer to?
In the Limitations, Conclusions and References sections:
- The first two sentences of the limitations section are repeated, even if they are written differently.
- The authors should rewrite the limitations section to make it clearer
- The conclusions must expand them
- The references section does not comply with the regulations of the journal, so the authors must review and modify it
Greetings
Author Response
Dear Esteemed Reviewer,
Thank you very much for reviewing our paper.
We modified the manuscript as follow:
In the methodology section>
- The type of the study was written in line 122 cross-sectional survey
- We moved the ethical considerations to the end of the methodology section
- The data was collected using Google doc. We search for the websites of major IT companies in Romania (mostly Bucharest and Cluj as having the majority of IT workers in Romania) and we sent the link and a short description of the study to the email listed as a contact on the website. We asked them to share the link on the company’s internal groups. Also, we sent the link directly to friends working in it with the request to share our link. (line 123)
- The sample was obtained via a snowball method, we email the link to the companies and people who first answered sent it also to colleagues.
In results.
- We corrected the font of the tables and adjusted them to be on the same page. From our knowledge, the journal makes the final formatting of the manuscript
- 1 we deleted the question mark -line 249
- We performed a regression analysis to see which factors influence the weight gain (the dependent variable) of computers workers. The factors that we took into consideration in our analysis were more hours of work in front of the computer, diet change, body mass index, gender, age. According to our results, more hours work in front of the computer, diet change, body mass index and age were associated with the weight gain of the computer workers (p<0.05). (line 326)
We deleted the dependent variable: weight gain, on line 333 (it was part of the statistical output and it is redundant as we have a title that explained the table. For the weight gain we had in a questionnaire a Yes/No question: did you gain weight during the pandemic.
In the Limitations, Conclusions and References section:
- We deleted a sentence, as recommended
- We modified the limitation. We added: Another limitation of our study was that we couldn′t reach the same sample of IT workers, during the pandemic as before the pandemic (the questionnaires were anonymous), but the two samples were comparable, without statistical differences regarding the age and sex distribution.(line 436-440))
- We expanded the conclusions: The study also revealed poor sleep quality among the majority of respondents, with at least one weeknight with less than 6 hours of sleep.
Regardless of the great number of working hours in front of computers, most IT workers perceived their stress related to works only as moderate. In addition, our study revealed that almost all workers suffered from social isolation during the pandemic, especially women (461-467).
- References were corrected according to the journal regulations. From our knowledge, the journal will correct them in the final formatting.
We sent the manuscript for Extensive editing of the English language, as recommended.
Kind regards,
Bogdana Nasui
Codruta Popescu

Reviewer 2 Report
Manuscript reports the effect of sedentary lifestyle induced by pandemic on the health and food intake parameters of a specific group of population- IT workers. Manuscript brings out the importance of balancing energy intake and output based on changes in the lifestyle.
Manuscript is written well but there is scope for improvement. English language needs correction at several places. There are several mistakes as to words missing, misspelled, or sentence not making appropriate sense.
Also, one concern about the analysis is the time duration between the participants. was it same across the boards or was it over a range of time.? Was the time factor taken into consideration when doing statistical testing.
There are some comments/suggestions in the attached file.

Author Response
Dear Esteemed Reviewer,
Thank you for reviewing our paper.
We corrected the sentences with mistakes, as recommended. Also, we corrected the English language using the Grammarly program.
We sent the manuscript for Extensive editing of the English language.
Please find in the revised attached manuscript all the corrections that we have made.
We added the table with the weight gain of the participants by BMI (Table 2), we obtained similar results when we used the percentage of body weight gained by categories.
We calculated the respondents’ weight gain depending on their body mass index. The results revealed that IT workers with overweight and obesity gained more weight than normo weight workers (Table2)
Table 2. The weight gain of the respondents depending of the BMI index
|
Obesity class (BMI) |
Mean± SD (kg) |
No (%) |
p-value |
|
Underweight |
5.83±5.49 |
8 (1.79) |
p<0.001 |
|
Normoweight |
5.85± 2.94 |
218 (48.80) |
|
|
Overweight |
6.81±3.56 |
170 (38.00) |
|
|
Obesity |
8.24±4.80 |
51 (11.41) |
*p<0,05 was considered statistically significant; ANOVA
The p-value from tables (e.g.table 1, line 22), showed the differences between all the BMI categories of males and females. We added the statistical test below the table.
Regarding the weight gain:
We consider the data collection as the first wave and second wave. We start sending the questionnaire on 27 October, got around 90% of the answers during the first 2 weeks of November, then we tried with friends and people we knew working in IT and got the rest of the questionnaire in December with the last ones in the first days of January. The answering pattern is typical for a study that used questionnaires shared electronically. You get the answers in the first two weeks since you sent the questionnaire, then very few new answers.
We performed some statistical calculations (t-test for the weight gain) for the people who gained weight to see if there were some statistical differences, and the results showed that there were no statistical differences that could be due to the 1-month differences between the data collection. Your suggestion is a very interesting hypothesis that could be validated in a much larger sample and a longer period for the data collection, but for our data, there are no statistical differences.
|
Weight gain |
Mean±SD (kg) |
No |
p-value |
|
Weight gain (first wave) |
5.24±3.54 |
414 |
0.413 |
|
Weight gain (second wave) |
4.72±2.77 |
33 |
Kind regards,
Bogdana Nasui
Codruta Popescu